# Robust and distributed neural representation of action values

**Eun Ju Shin[1,2†], Yunsil Jang[1,2†], Soyoun Kim[3], Hoseok Kim[4], Xinying Cai[5], Hyunjung Lee[6], Jung Hoon Sul[1], Sung-Hyun Lee[7], Yeonseung Chung[8], Daeyeol Lee[9]\*, Min Whan Jung[1,2]\***

[1]Center for Synaptic Brain Dysfunctions, Institute for Basic Science, Daejeon, Republic of Korea; [2]Department of Biological Sciences, Korea Advanced Institute of Science and Technology, Daejeon, Republic of Korea; [3]Center for Neuroscience Imaging Research, Institute for Basic Science, Suwon, Republic of Korea; [4]Department of Neuroscience, Biomedicum, Karolinska Institutet, Stockholm, Sweden; [5]New York University Shanghai, NYU-ECNU Institute of Brain and Cognitive Science at NYU Shanghai, and Shanghai Key Laboratory of Brain Functional Genomics (Ministry of Education), School of Psychology and Cognitive Science, East China Normal University, Shanghai, China; [6]Department of Anatomy, Kyungpook National University School of Medicine, Daegu, Republic of Korea; [7]Neuroscience Graduate Program, Ajou University School of Medicine, Suwon, Republic of Korea; [8]Department of Mathematical Sciences, Korea Advanced Institute of Science and Technology, Daejeon, Republic of Korea; [9]The Zanvyl Krieger Mind/Brain Institute, Kavli Neuroscience Discovery Institute, Department of Neuroscience, and Department of Psychological and Brain Sciences, Johns Hopkins University, Baltimore, United States

**\*For correspondence:**
daeyeol@jhu.edu (DL);
mwjung@kaist.ac.kr (MWJ)

[†]These authors contributed equally to this work

**Abstract** Studies in rats, monkeys, and humans have found action-value signals in multiple regions of the brain. These findings suggest that action-value signals encoded in these brain structures bias choices toward higher expected rewards. However, previous estimates of action-value signals might have been inflated by serial correlations in neural activity and also by activity related to other decision variables. Here, we applied several statistical tests based on permutation and surrogate data to analyze neural activity recorded from the striatum, frontal cortex, and hippocampus. The results show that previously identified action-value signals in these brain areas cannot be entirely accounted for by concurrent serial correlations in neural activity and action value. We also found that neural activity related to action value is intermixed with signals related to other decision variables. Our findings provide strong evidence for broadly distributed neural signals related to action value throughout the brain.

## Introduction

The reinforcement learning theory provides a general theoretical framework for understanding the neural basis of value-based decision making (*Corrado and Doya, 2007*; *Dayan and Niv, 2008*; *Glimcher, 2011*; *Lee et al., 2012a*; *Mars et al., 2012*; *O'Doherty et al., 2007*). In algorithms based on this theory, an agent selects an action based on a set of action values (i.e., values associated with potential actions) in a given state (*Sutton and Barto, 1998*). Human and animal choice behaviors are parsimoniously accounted for by such algorithms. Furthermore, a large body of studies in rats, monkeys, and humans have found neural or hemodynamic signals correlated with action value in multiple regions of the brain, especially in the frontal cortex-basal ganglia loop (*Chase et al., 2015*; *Ito and*

*Doya, 2011*; *Lee, 2006*; *Lee et al., 2012a*; *Rushworth et al., 2009*). These findings led to the view that multiple brain structures contribute to biasing choices toward relatively valuable ones during decision making by representing a set of action values.

It is often difficult to rigorously demonstrate that neural activity is genuinely correlated with action value, because both neural activity and action value tend to fluctuate slowly over time and thus are serially correlated. Recently, for example, *Elber-Dorozko and Loewenstein, 2018* made two lines of argument to suggest that action-value neurons had not been clearly demonstrated in the striatum. First, with a permutation test in which behavioral data from different sessions are used to determine significance of action-value-related neural activity, they found that the number of neurons encoding action value in the ventral striatum (VS) and ventral pallidum (VP; *Ito and Doya, 2009*) was reduced to a chance level. A more recent simulation study also has shown that naïve applications of the conventional F-test for multiple linear regression can suffer from a false positive and hence a 'nonsense correlation' between a behavioral variable and autocorrelated neural activity (*Harris, 2020*). Second, *Elber-Dorozko and Loewenstein, 2018* argued that neural activity related to action value may reflect other decision variables correlated with action value, such as a choice probability or policy. Even though *Elber-Dorozko and Loewenstein, 2018* focused on striatal action-value signals, these problems might be also relevant to action-value signals reported in other brain areas.

Given the significance of these statistical issues concerning value-related signals throughout the brain, we decided to reanalyze the data we have collected in our previous studies using the methods designed to strictly account for temporal correlations in the data. In addition to the permutation test used in *Elber-Dorozko and Loewenstein, 2018*, we also used surrogate behavioral and neural data to determine the statistical significance of value signals. We also tested whether action-value neurons identified in our previous studies merely encode policy or state value rather than action value. Overall, the results from these analyses demonstrate that neural activity in many areas of the brain, including the striatum, robustly encode action values.

## Results

### Neuronal and behavioral database

We analyzed neural activity related to action value as well as chosen value (value of the chosen action in a given trial). Included in this analysis are the neural data recorded from the dorsomedial striatum (DMS, 466 neurons), dorsolateral striatum (DLS, 206 neurons), VS (165 neurons), lateral orbitofrontal cortex (OFC, 1148 neurons), anterior cingulate cortex (ACC, 673 neurons), medial prefrontal cortex (mPFC, 854 neurons), secondary motor cortex (M2, 411 neurons), and dorsal CA1 region (508 neurons) in rats (n = 27; 383 sessions) performing a dynamic foraging task in a modified T-maze (*Figure 1*, see Materials and methods; *Kim et al., 2009*; *Kim et al., 2013*; *Sul et al., 2010*; *Sul et al., 2011*; *Lee et al., 2012b*; *Lee et al., 2017*). We also analyzed neural data recorded from the dorsolateral prefrontal cortex (DLPFC, 164 neurons), caudate nucleus (CD, 93 neurons), and VS (90 neurons) in three monkeys performing an intertemporal choice task (see Materials and methods; *Kim et al., 2008*; *Cai et al., 2011*). In these monkey experiments, temporally discounted values (DVs) of alternative choices were randomized across trials, so that all decision variables were devoid of temporal correlation. We included in the analysis only those neurons with mean firing rates ≥1 Hz during a given analysis time window. To assess action-value-related neural activity in rats, we analyzed neural spike data during the last 2 s of the delay period, immediately before the central bridge is lowered so that the animal is allowed to run forward and head toward the left or right goal location (*Figure 1A*; 196 DMS, 123 DLS, 68 VS, 782 OFC, 405 ACC, 431 mPFC, 301 M2, and 307 CA1 neurons). To assess action-value-related neural activity in monkeys, we analyzed neural spike data during the 1 s time window before the onset of sensory cues signaling two choice options (75 CD, 66 VS, and 105 DLPFC neurons). To assess chosen-value-related neural activity in rats, we analyzed neural spike data during the 2 s time period centered around the outcome onset (±1 s since the choice outcome was revealed; 241 DMS, 139 DLS, 80 VS, 808 OFC, 401 ACC, 446 mPFC, 334 M2, and 326 CA1 neurons). In the following, we first describe the results from simulations to test false positive rates of several different statistical tests used in the present study in identifying action-value and chosen-value neurons. We then show the results of these tests applied to the actual neural data collected from rats performing the block-designed dynamic foraging task. We then address the

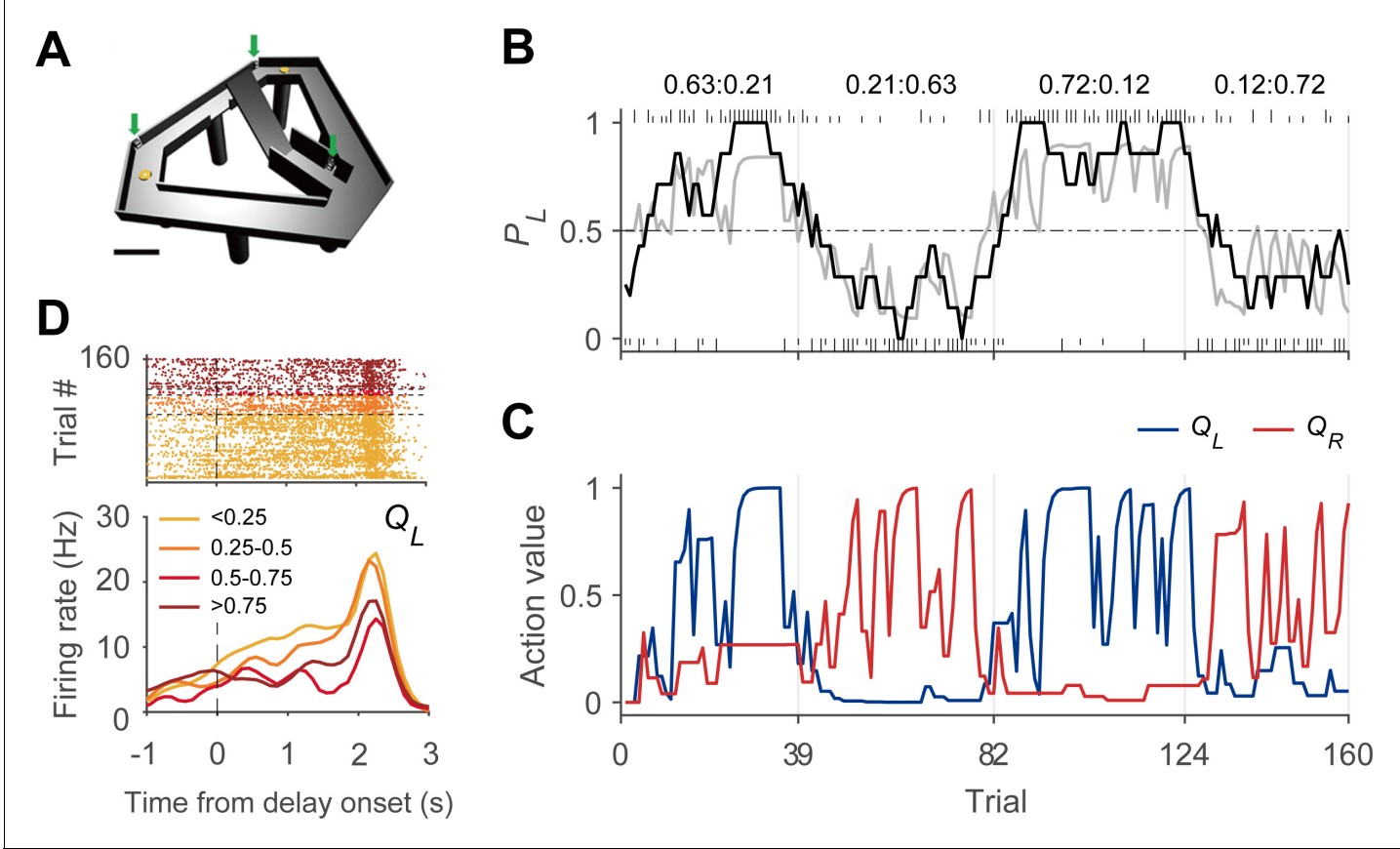

**Figure 1.** Dynamic foraging task. (**A**) Modified T-maze. Rats chose freely between two targets (orange circles) to obtain water reward. Rats navigated from the central stem to either target and returned to the central stem via the lateral alley to start a new trial. A delay (2–3 s) was imposed at the beginning of a new trial by raising the central bridge. Green arrows, photobeam sensors. Scale bar, 10 cm. (**B**) Behavioral data from a sample session (*Kim et al., 2013*). The black curve shows the probability to choose the left target ($P_L$) in moving average of 10 trials. The gray curve denotes the probability to choose the left target predicted by the Q-learning model. Tick marks denote trial-by-trial choices of the rat (upper, left choice; lower, right choice; long, rewarded trial; short, unrewarded trial). Vertical gray lines denote block transitions and numbers above indicate reward probabilities of the left and right targets in each block. (**C**) Trial-by-trial action values of the sample session computed with the Q-learning model. Blue, left-choice action value ($Q_L$); Red, right-choice action value ($Q_R$). (**D**) An example DMS unit showing activity correlated with left-choice action value. Trials were grouped into quartiles of left-choice action value. Delay onset is when the rat broke the photobeam sensor on the central stem.

issue of potentially misidentifying other decision-variable signals as action-value signals using the data from both rats and monkeys.

## Validation of permutation and surrogate data-based tests

We first assessed false positive rates of different analysis methods using actual behavioral data and simulated null neural data whose autocorrelation was chosen to match that of the actual neural data. The simulated neural data was generated as the following:

$$x(t) = \alpha \cdot x(t-1) + \epsilon, \tag{1}$$

where *x(t)* is the firing rate of the simulated neuron at trial *t*, $\alpha$ is the autoregressive (AR) coefficient, and ε is a standard normal deviate. We then generated time series for spike counts assuming the simulated neuron is a Poisson process. We set $\alpha$ = 0.8 and 0.83 to test false positive rates of action-value and chosen-value signals, respectively, which were chosen to match the distributions of the first-order AR coefficient, AR(1), and mean firing rate to those of the actual neural data used to analyze action-value signals (neural activity during the last 2 s of the delay period; AR(1) = 0.19 ± 0.18 and mean firing rate = 6.14 ± 7.61 Hz, n = 2613 neurons) and chosen-value signals (neural activity

during the 2 s time period centered around the outcome onset; AR(1) = 0.21±0.20 and mean firing rate = 5.90 ± 6.72 Hz, n = 2775 neurons; mean ± SD).

We used these simulated neural data to test false positive rates of different analysis methods. Throughout the study, we identified action-value neurons as those whose activity is significantly related to either of the left and right action values (p<0.025 for $Q_L$ or $Q_R$). A conventional *t*-test (linear regression analysis, model 1, *Equation 5*) yielded >10% action-value neurons, which is significantly greater than expected by chance (binomial test, p=3.7 × $10^{-22}$). Adding potentially confounding variables to the regression (choice and chosen value; model 2, *Equation 6*) reduced the number of action-value neurons, but it was still significantly greater than expected by chance (binomial test, p=2.6 × $10^{-9}$; *Figure 2A*). We used two different methods to handle false positive identification of action-value neurons in our previous studies. One method (within-block permutation; see Materials and methods; *Kim et al., 2009*) reduced the false positive rate further, but it was still significantly above the chance level (binomial test, p=5.9 × $10^{-5}$). The other method (adding AR terms to the regression; see Materials and methods; *Kim et al., 2013*; *Sul et al., 2010*; *Sul et al., 2011*; *Lee et al., 2012b*; *Lee et al., 2017*) reduced the number of action-value neurons to the chance level (binomial test, p=0.191; *Figure 2A*). The use of the same tests was less problematic for the analysis of chosen-value signals. A conventional *t*-test (model 4, *Equation 8*) yielded ~9% chosen-value neurons and it was significantly greater than expected by chance (binomial test, p=3.7 × $10^{-8}$). However, the number of chosen-value neurons was reduced to the chance level by adding confounding variables to the regression (model 5, *Equation 9*) and also by other methods used in our previous studies (applying within-block permutation or adding AR terms to model 5; *Figure 2B*).

We then tested four different methods based on surrogate data to determine statistical significance of value terms in multiple regression models (models 2 and 5; *Equation 6 and 9*). The first two methods used surrogate behavioral data. Specifically, we tested session permutation (*Elber-Dorozko and Loewenstein, 2018*) and pseudosession (*Harris, 2020*) methods. In the session permutation test, surrogate behavioral data was drawn from other behavioral sessions. In the pseudosession test, surrogate behavioral data for a given session was generated based on a reinforcement learning model using the model parameters estimated for the same animal (see Materials and methods). The other two methods used surrogate neural data generated with Fourier phase randomization (FPR). For this, we tested the conventional FPR method and the amplitude adjusted Fourier transformation (AAFT) method (*Theiler et al., 1992*; see Materials and methods). Both methods generate surrogate neural data with the same amplitude, but randomized phase of the Fourier transform as the original data. The two methods differ in that the surrogate neural data has a normalized spike count distribution (FPR) or maintains the original spike count distribution (AAFT). In all of these methods, the p-value for a regression coefficient was determined by the frequency in which the magnitude of *t*-value obtained using surrogate data exceeds that of the original *t*-value. When tested using the simulated neural data, all of these four methods yielded ~5% of false positive action-value and chosen-value neurons, and none of them was significantly higher than expected by chance (binomial test, p>0.05; *Figure 2A,B*) Therefore, these tests are unlikely to suffer from an inflated false positive rate when applied to our actual neural data.

For the session permutation method, we found that trial-by-trial action values are substantially correlated between the original and resampled behavioral sessions. We used four different combinations of reward probabilities (left:right = 0.72:0.12, 0.63:0.21, 0.21:0.63 and 0.12:0.72) in our previous studies and, even though their sequence was randomized, there was a constraint that the option with the higher-reward probability always changes its location at the beginning of a new block. The number of trials per block was also similar across studies (40.1 ± 3.1; mean ± SD). Hence, the relative reward probability tended to be correlated or anti-correlated between two different sessions depending on whether the first blocks of the two sessions had the same or different locations for the higher-reward-probability target (*Figure 2C*). Likewise, in the pseudosession method, which generates simulated behavioral data according to the same block structure of a given behavioral session, trial-by-trial action values tended to be positively correlated between actual and simulated behavioral sessions (*Figure 2C*). This raises the possibility that for the neural data collected during the experiments with a block design, the session permutation and pseudosession methods might be too stringent (high false negative rate) for the identification of action-value neurons. Unlike action

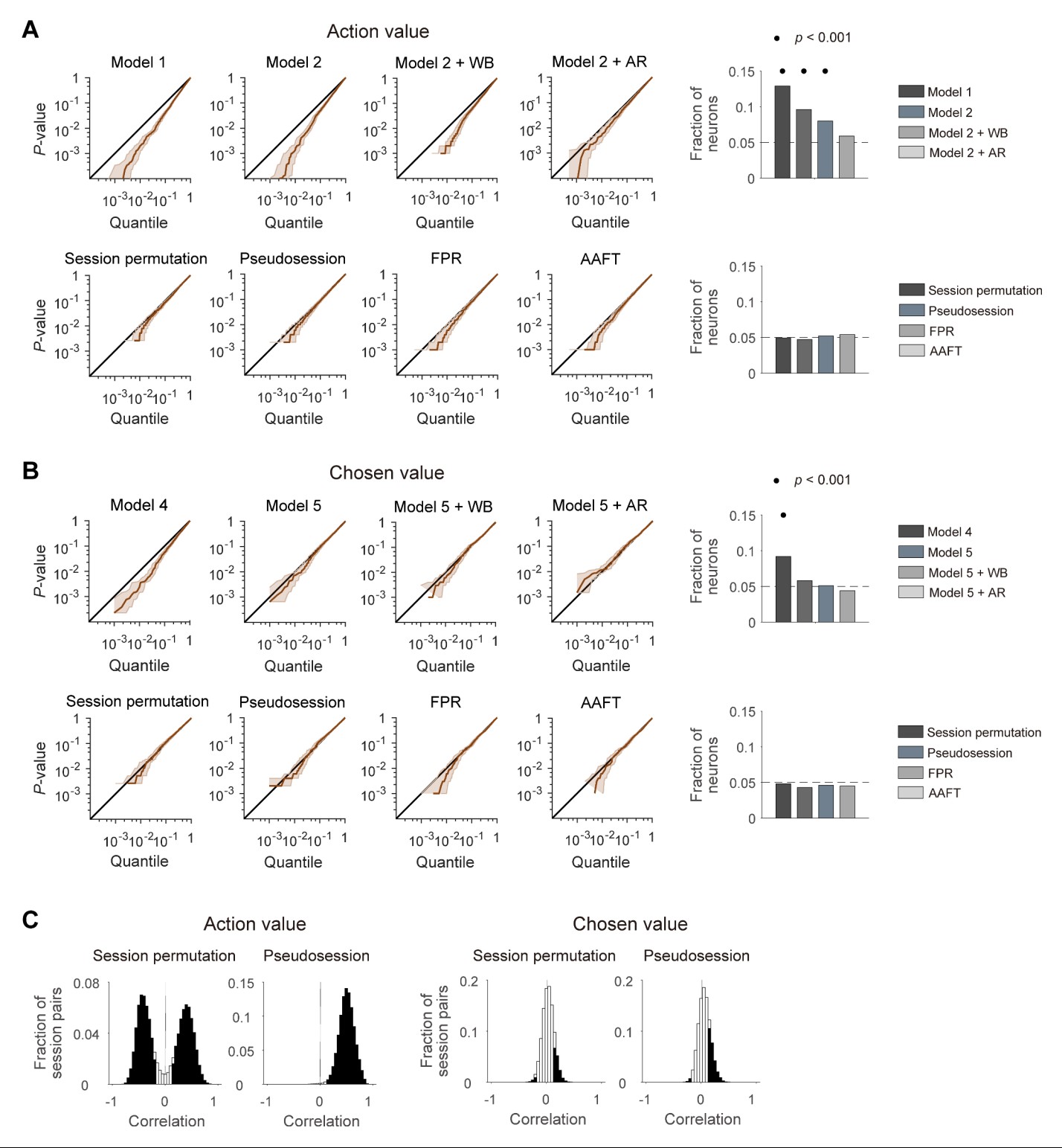

**Figure 2.** Performances of different statistical tests for action-value and chosen-value signals. (**A**) Left, cumulative density functions (CDFs) of p-values for the neural activity related to action value were determined with different analysis methods using null neural data to assess false positive rates. Results obtained with the methods used in previous studies (top), including the *t*-test in two different regression models (models 1 and 2), within-block permutation applied to model 2 (model 2 + WB), and model 2 with autoregressive terms (model 2 + AR), as well as those obtained with the methods based on surrogate data (bottom) are shown. Right, fractions of neurons significantly responsive to either action value (p<0.025 for $Q_L$ or $Q_R$). Horizontal dotted lines denote 5%. Significant fractions (binomial test) are indicated by black filled circles. (**B**) Left, CDFs of p-values for the neural

*Figure 2 continued on next page*

*Figure 2 continued*

activity related to chosen value. Right, fractions of neurons significantly responsive to chosen value (p<0.05 for $Q_c$). The same format as in **B**, but models 4 and 5 were used instead of models 1 and 2, respectively. (**C**) Correlation between action values (left) or chosen values (right) calculated from the original and resampled behavioral data either from other sessions (session permutation, n = 382) or simulated behavioral sessions (pseudosession, n = 500). Filled bars indicate significant (*t*-test, p<0.05) correlations.

values, trial-by-trial chosen values were only weakly correlated between the original and resampled behavioral sessions (*Figure 2C*).

## Activity related to action value and chosen value

We applied the above methods to the actual neural data obtained from rats. We analyzed the neural data during the last 2 s of the delay period to assess action-value-related neural activity. As expected, the conventional *t*-tests yielded high levels of action-value signals and they were reduced substantially by employing the within-block permutation procedure or adding AR terms. All of these methods yielded significant (binomial test, p<0.05) fractions of action-value neurons in all tested brain structures except the DLS (*Figure 3A*, top). The pseudosession, FPR, and AAFT methods also yielded significant action-value signals in all of these brain structures except the DLS. The proportion of action value-coding neurons tended to be somewhat lower when they were determined with the session permutation method, but this was still significantly above the chance level in several brain areas, including the striatum, OFC, and hippocampus (*Figure 3A*, bottom). When applied to neural data during the 2 s time period centered around the outcome onset, all of these methods yielded significant chosen-value signals in all tested brain structures (*Figure 3B*). These results show significant encoding of action-value and chosen-value signals in multiple areas of the rat brain that cannot be explained by slowly drifting and serially correlated neural activity.

## Transformation of value signals

In reinforcement learning theory, action values are monotonically related to the probability of choosing the corresponding actions, referred to as policy, making it hard to distinguish the neural activity related to either of these quantities. In addition, the activity of individual neurons is likely to encode multiple variables simultaneously (*Rigotti et al., 2013*). Despite these difficulties, it has been argued that neural signals related to action value might actually represent policy exclusively (*Elber-Dorozko and Loewenstein, 2018*). To address this issue quantitatively, we used the difference in action values ($\Delta Q$) and their sum ($\Sigma Q$) as proxies for policy and state value, respectively, and tested how signals for action value, policy, and state value are related in a population of neurons in different brain structures.

If the activity of a given neuron is strongly related to policy, then its activity would be related to the difference in action values, $\Delta Q$, but not their sum, $\Sigma Q$. To test whether this is the case, we analyzed the same neural data used to assess action-value-related neural activity in rats (neural spikes during the last 2 s of the delay period). As shown in *Figure 4B*, some of the action value-responsive neurons showed activity correlated with $\Sigma Q$ (25.8–62.5% across different brain areas), some with $\Delta Q$ (13.2–42.9%), and others with both $\Sigma Q$ and $\Delta Q$ (6.3–25%). There were also neurons that were exclusively responsive to action value (0–22.7%). Conversely, some of $\Sigma Q$- and $\Delta Q$-responsive neurons were also responsive to action value ($\Sigma Q$, 59.1–100%; $\Delta Q$, 11.5–38.5%) and some were exclusively responsive to $\Sigma Q$ (0–40.9%) or $\Delta Q$ (61.5–88.5%). These results indicate that a population of neurons in many brain areas tend to represent all of these variables rather than exclusively representing only one type of decision variable. For comparison, we also analyzed neural activity recorded during the 1 s time window before cue onset from the CD (a part of the DS), VS, and DLPFC of monkeys performing an intertemporal choice task (*Cai et al., 2011*; *Kim et al., 2008*). The results from this analysis were similar to those obtained from rats (*Figure 4C*), suggesting that DLPFC and striatal neurons in monkeys also represent all of these variables rather than exclusively representing only one type of value signals.

Even though all the brain regions tested in this study represented multiple types of value signals in parallel, their relative signal strengths varied across brain regions. If multiple types of value signals are represented equally often and strongly, then the points in *Figure 4* would be rotationally invariant. By contrast, the pattern of anisotropy in these plots would change if the neurons in a given brain

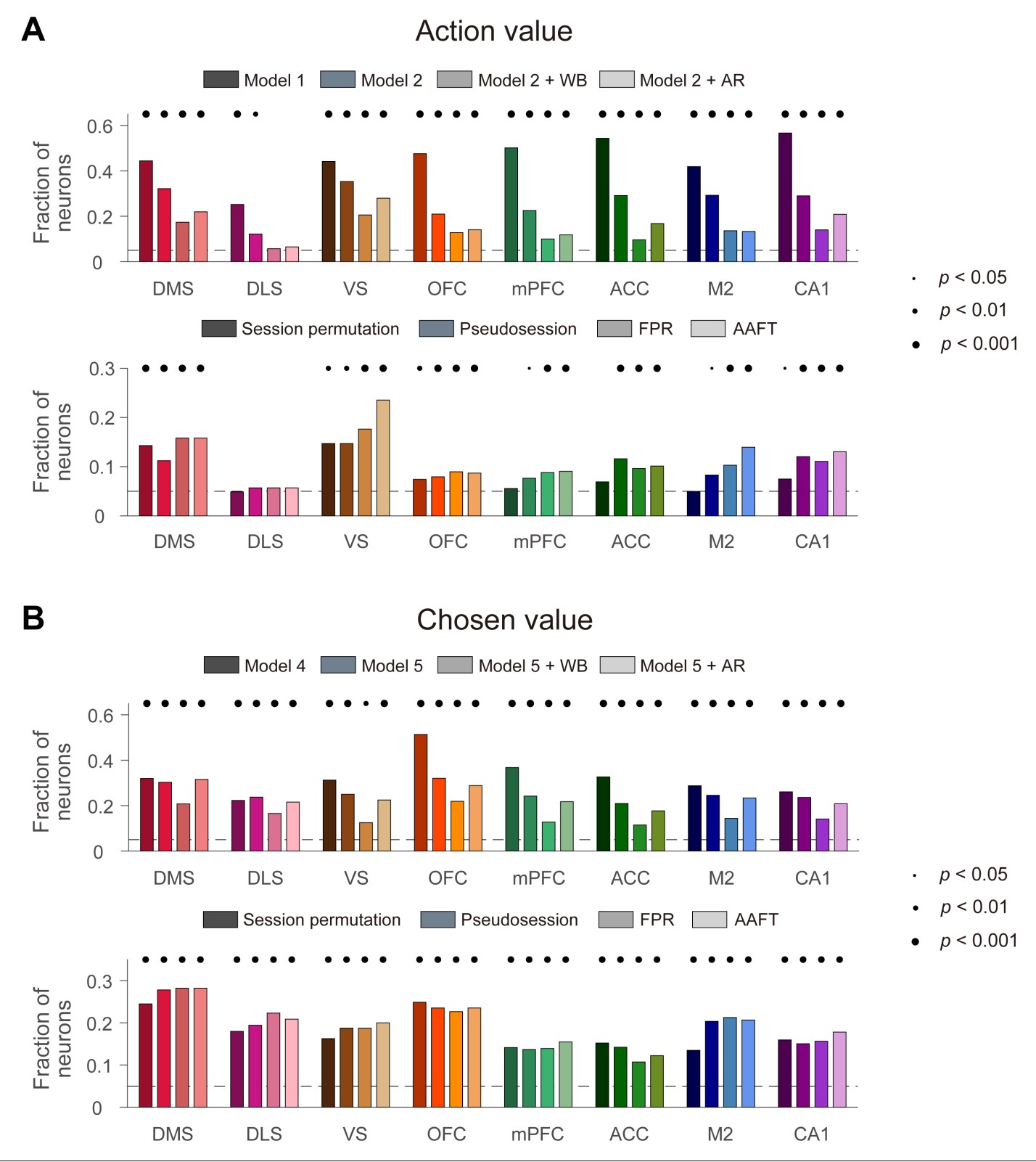

**Figure 3.** Action-value and chosen-value signals in multiple brain regions. Action-value and chosen-value neurons were determined based on actual behavioral data and actual neural data recorded from several different areas of the rat brain. Shown are fractions of neurons significantly responsive to either action value ($p<0.025$ for $Q_L$ or $Q_R$; **A**) or chosen value ($p<0.05$ for $Q_c$; **B**) determined with the previous methods (top) or resampling-based methods (bottom). Significant fractions (binomial test) are indicated by black filled circles.

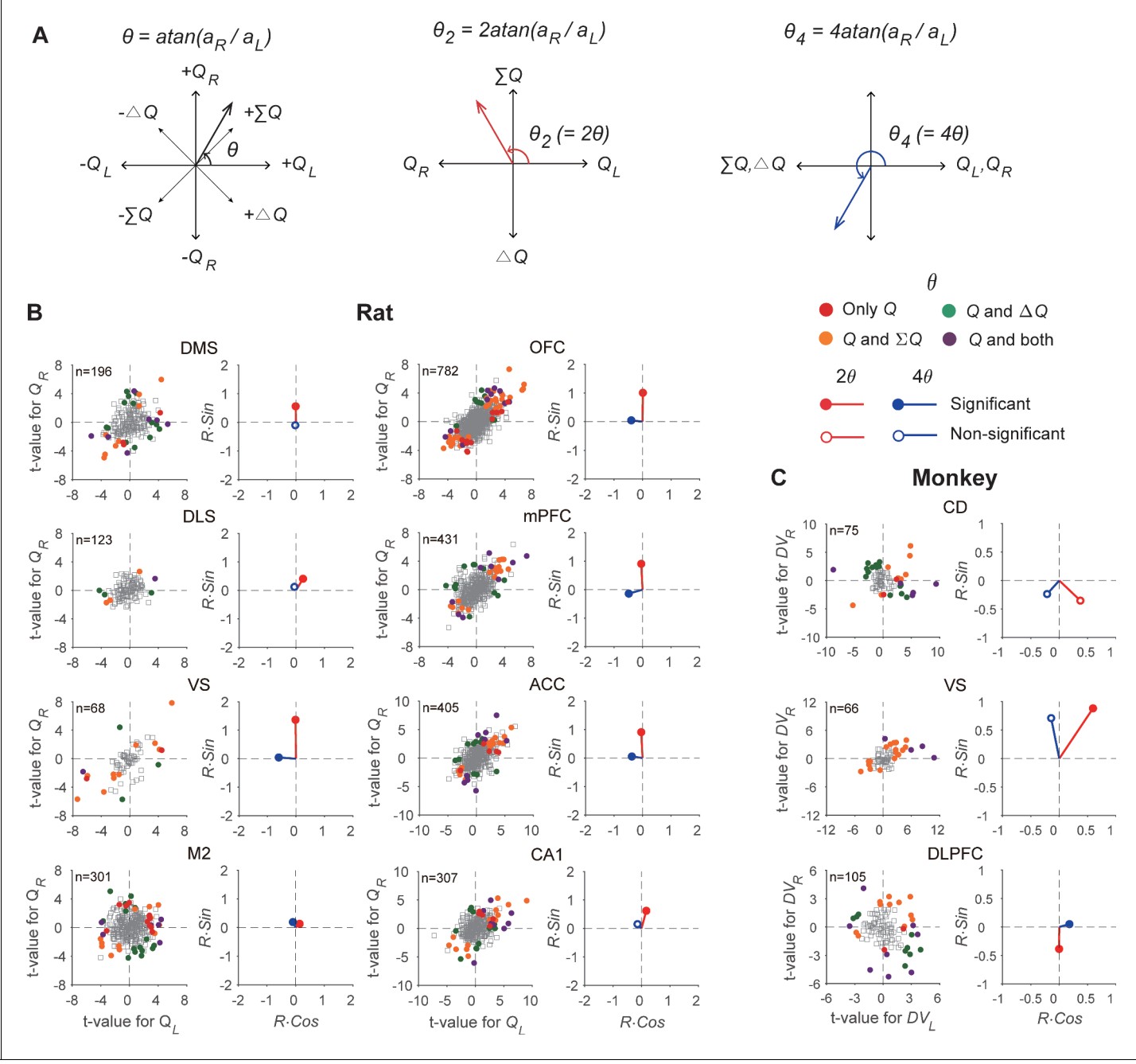

**Figure 4.** Neural signals related to action value, policy, and state value. (**A**) Transformations applied to the angle defined by the original regression coefficients ($\theta$) to examine multiple types of value signals ($\theta_2$ for signals related to policy vs. state value; $\theta_4$ for action values vs. other value signals). (**B**) The scatter plots show *t*-values for the left and right action values (abscissa and ordinate, respectively) estimated from neural activity recorded in different areas of the rat brain. Filled circles denote those neurons significantly responsive to one or more of the decision variables tested ($Q_L$, $Q_R$, $\Delta Q$, and $\Sigma Q$). *Q*, those neurons significantly responsive to either action value ($p<0.025$ for $Q_L$ or $Q_R$). The vectors on the right panel for each area show mean vectors computed after doubling ($2\theta$, red) or quadrupling ($4\theta$, blue) the angle of each data point in the scatter plots. Red filled circles, the Y-component of the mean vector is significantly different from 0; blue filled circles, the X-component of the mean vector is significantly different from 0 (Wilcoxon rank-sum test, $p<0.05$). (**C**) The same scatter and vector plots for monkey striatal and DLPFC neurons. $DV_L$, left action value; $DV_R$, right action value.

area tend to encode a specific type of value signals. For example, if neurons in a given brain area mostly encode $\Sigma Q$, the points representing individual neurons would be clustered along the identity line, since the regression coefficients for $Q_L$ and $Q_R$ would be similar for such neurons (*Figure 4A*). As shown in *Figure 4*, the distribution pattern of $Q_L$-versus-$Q_R$ regression coefficients varied substantially across regions.

To quantify this further, we computed the mean resultant vectors after multiplying the angle of the vector defined by the regression coefficients for action values, θ (see Materials and methods), by a specific factor. First, we compared the vertical component of the mean resultant vector calculated after doubling these angles to test whether neurons in each area might be biased for coding $\Sigma Q$ or $\Delta Q$ (*Figure 4A*). The results from this analysis showed that the vertical component of the mean resultant vector was significantly positive in all regions in the rat (Wilcoxon rank-sum test, statistical test results summarized in *Supplementary file 1*; see also *Figure 4B*), indicating stronger encoding of $\Sigma Q$ than $\Delta Q$ signals. In addition, the vertical component of this resultant varied in magnitude across regions. In the striatum, it was significantly larger in the VS than in the DMS and DLS. In the cortical areas, it was significantly larger in the OFC, mPFC, and ACC than in the M2 (one-way ANOVA followed by Bonferroni post hoc tests, statistical test results summarized in *Supplementary file 1*). In the monkey, the vertical component of the mean resultant vector was significantly negative in the DLPFC (Wilcoxon rank-sum test; *Figure 4C*), indicating stronger encoding of $\Delta DV$ than $\Sigma DV$ signals in the DLPFC. In addition, the vertical component of the resultant in the VS was significantly different from those in the DLPFC and CD, suggesting that the VS neurons tended to encode $\Sigma DV$ signals more strongly than DLPFC and CD neurons (statistical test results summarized in *Supplementary file 1*). Collectively, these results showed that $\Sigma Q$ signals are generally stronger than $\Delta Q$ signals in the rat brain areas examined in this study, and $\Delta DV$ signals are particularly strong in the monkey DLPFC.

Next, we examined the horizontal component of the mean resultant vector after multiplying the angles of the regression coefficient vector by four in order to test whether neurons in each brain area tend to favor coding action values of individual choices or whether they tend to combine action values to encode policy or state value. After this transformation, $Q_L$- and $Q_R$-coding neurons together would form vectors along the X-axis with positive horizontal components, whereas $\Sigma Q$- and $\Delta Q$-coding neurons together would form vectors along the X-axis in the negative domain (*Figure 4A*). The results from this analysis showed that the horizontal component of the mean resultant vector was significantly negative in the VS, OFC, mPFC, ACC, and M2, but not in the other regions of the rat brain (Wilcoxon rank-sum test; *Figure 4B and C*; statistical test results summarized in *Supplementary file 1*), indicating that signals related to $\Sigma Q$ and $\Delta Q$ were more strongly represented than action-value signals of individual choices in the rat VS, OFC, mPFC, ACC, and M2.

## Discussion

Neural signals related to action value have been found in widespread regions of the brain, especially in the frontal cortex-basal ganglia loop (*Chase et al., 2015*; *Ito and Doya, 2011*; *Lee, 2006*; *Lee et al., 2012a*; *Rushworth et al., 2009*), suggesting the involvement of multiple brain structures in value-based decision making. However, the potential confounding of concurrent autocorrelations in value signals and neural activity and the possible superposition of different types of value signals have not been clearly resolved. The results in the present study confirm significant action-value signals in most of the brain regions tested previously. We also found that action-value-related neural activity coexists with that related to policy and state values. These results support previous conclusions that action values are represented in many regions of the brain. Below, we discuss these two issues along with the significance of anatomical variation in value signals.

### Concurrence of autocorrelation in behavioral and neural data

Neural spikes are often correlated across trials, as we demonstrated for all brain structures examined in the present study. Note that serial correlation in neural activity could be due to intrinsic non-stationarity and/or its relationship with action value. In the present study, we reanalyzed our previous neural data to rigorously test whether action-value neurons identified in previous block-design studies might result from serial correlation in neural spikes unrelated to action value. Recently, this issue was examined with simulated neural and behavioral data (*Elber-Dorozko and Loewenstein, 2018*;

*Harris, 2020*), but the nature of serial correlations in simulated neural spikes and action value might deviate substantially from those of actual neural and behavioral data. In the present study, using actual behavioral data and simulated neural data whose level of autocorrelation was matched to that of the actual neural data, we first established that four different analysis methods, namely the session permutation, pseudosession, FPR and AAFT methods (*Elber-Dorozko and Loewenstein, 2018*; *Harris, 2020*; *Theiler et al., 1992*), do not inflate action-value signals. Applying these methods to actual neural data, we still found significant action-value signals in multiple areas of the rat brain. These findings indicate that action-value signals in our previous block-design studies cannot be entirely attributed to concurring autocorrelations in behavioral data and neural spikes unrelated to action value.

It should be noted that the optimal methods to assess value-related neural activity might vary depending on exact structures of neural and behavioral data. In our studies, because of similarity in block structure across sessions, trial-by-trial action values were substantially correlated across sessions. This suggests that session permutation might be excessively stringent for testing action-value-related neural activity. This is similarly problematic for the pseudosession test because simulated behavioral sessions have the same block structure as the original behavioral session. Indeed, both methods yielded somewhat lower fractions of action-value neurons compared to the FPR and AAFT tests. The pseudosession method yielded somewhat higher fractions of action-value neurons than the session permutation test in most tested regions, which suggests that some variability in the animal's behavior shared across different sessions (e.g., a slow change in motivation) might not be captured by the models used to estimate action values. For blocked behavioral sessions, therefore, the FPR and AAFT methods are likely to estimate action-value signals more accurately than the session permutation and pseudosession methods. Our results also suggest that the problem arising from serial correlation in neural activity can be ameliorated by adding AR terms in the regression model. In our study, the results obtained with the FPR and AAFT methods were similar. Nevertheless, the simulated neural data obtained with the FPR lose their discrete properties and become normally distributed, whereas the AAFT maintains the original distribution of spike counts. Therefore, the AAFT method might be more reliable when the neural signals of interest are influenced by a non-Gaussian or discrete nature of neural data. Neural activity is almost always serially correlated, and this makes it difficult to select appropriate statistical methods to identify how sensory, motor, or other cognitive variables are encoded in the brain when they are also serially correlated. For each candidate analysis method, therefore, it would be prudent to examine the rates of potential false positivity and negativity using a null data set that captures important features of the data set under investigation.

## Multiple types of value signals

Neural activity seemingly representing action value might in fact represent other decision variables, such as policy, that are correlated with action value (*Elber-Dorozko and Loewenstein, 2018*). To test this, we compared neuronal responses to action value with those related to the sum of two action values and their difference as proxies for neuronal responses related to state value and policy, respectively. We found neurons carrying diverse combinations of value-related signals in the striatum, frontal cortical areas, and hippocampus. The majority of action-value coding neurons also coded state value and/or policy and, conversely, the majority of state value- and/or policy-coding neurons also coded action value as well. Also, a small number of neurons encoded action value without state value or policy, and some neurons encoded state value or policy without action value. Similarly, using a task in which values associated with specific colors and locations of sensory cues can be dissociated, we have shown previously that partially overlapping populations of neurons represent values associated with target colors and locations in the striatum and DLPFC in monkeys (*Kim et al., 2012*). Collectively, these results suggest that neurons in the striatum, frontal cortical areas, and hippocampus might not represent multiple types of value signals categorically, but instead show random mixed selectivity (*Hirokawa et al., 2019*; *Raposo et al., 2014*). Namely, the results from our analyses suggest that relatively weights given to different types of value signals vary continuously across individual neurons in most brain areas.

In addition to this heterogeneity in value coding within each brain region, how different types of value signals are encoded by individual neurons also varied across the brain structures examined in the present study. In the rat, all the tested regions, especially the OFC, mPFC, ACC, and VS, tended to over-represent signals related to the sum of action values, but this tendency was weaker in the

M2. These results suggest that the OFC, mPFC, ACC, and VS might mainly process signals related to the expected rewards that can be obtained in a given state (*Bari et al., 2019*), whereas the M2 might be concerned more with policy and action selection (*Sul et al., 2011*). The primate DLPFC conveyed relatively strong signals correlated with the difference between action values, suggesting its function might be more strongly related to policy and action selection than state value. These findings are at odds with functional homology between the rodent mPFC and monkey DLPFC (*Uylings et al., 2003*; *Vertes, 2006*). As in the rat striatum, we found a stronger representation of signals related to the sum of action values in the VS than in the CD in monkeys. This is consistent with the proposal that subdivisions of the striatum correspond to distinct cortico-basal ganglia loops serving different functions (*Devan et al., 2011*; *Ito and Doya, 2011*; *Redgrave et al., 2010*; *Yin and Knowlton, 2006*). Further studies are needed to clarify relative strengths of different decision variables in different brain structures and how they are related to the functions served by individual brain structures.

## Materials and methods

### Behavioral and neural data

We analyzed single-neuron activity recorded from the dorsomedial (DMS, n = 466), dorsolateral (DLS, n = 206), and ventral (VS, n = 165) striatum of six rats performing a dynamic foraging task (a total of 81 sessions) in our previous studies (*Kim et al., 2013*; *Kim et al., 2009*), as well as activity recorded from the lateral OFC (n = 1148, three rats), ACC (n = 673, five rats), mPFC (n = 854, six rats), M2 (n = 411, three rats), and dorsal CA1 (n = 508, 11 rats) in our previous studies (total 302 sessions; *Sul et al., 2010*; *Sul et al., 2011*; *Lee et al., 2012b*; *Lee et al., 2017*). For the analysis of action-value signals, we focused on neural activity during the last 2 s interval of the delay period and included only the neurons with mean discharge rates ≥1 Hz during the analysis window. For the analysis of chosen-value signals, we analyzed the activity during the 2 s time period centered around the outcome onset for the neurons with mean discharge rates ≥1 Hz during the analysis window. We also analyzed neural activity previously recorded in the CD, VS, and DLPFC of three monkeys performing an intertemporal choice task (*Cai et al., 2011*; *Kim et al., 2008*). This analysis was based on the activity during the 1 s cue period of the neurons with mean discharge rates ≥1 Hz.

### Behavioral task

Details of behavioral tasks have been published previously (*Cai et al., 2011*; *Kim et al., 2008*; *Kim et al., 2013*; *Kim et al., 2009*; *Lee et al., 2012a*; *Lee et al., 2017*; *Sul et al., 2011*; *Sul et al., 2010*). Briefly, each rat performed one of two different dynamic foraging tasks. Each trial began as the rat returned to the central stem (detected by a photobeam sensor; green arrow in *Figure 1A*) of a modified T-maze from either target location (orange circles in *Figure 1A*). After a delay of 2–3 s, the central bridge was lowered (delay offset) allowing the rat to navigate forward and choose freely between the two goal locations to obtain water reward. The rats performed four blocks of trials with each block associated with one of four different reward probability pairs (left:right = 0.72:0.12, 0.63:0.21, 0.21:0.63 or 0.12:0.72). The sequence of block was randomly determined with the constraint that the higher-probability target changes its location at the beginning of each block. In the two-armed bandit (TAB) task (n = 215 sessions, n = 17 rats; *Kim et al., 2009*; *Lee et al., 2012b*; *Sul et al., 2010*), water was delivered probabilistically only at the chosen location in a given trial, whereas in the dual assignment with hold (DAWH) task (n = 168 sessions, n = 10 rats; *Kim et al., 2013*; *Lee et al., 2017*; *Sul et al., 2011*), water was delivered probabilistically at both locations according to a concurrent variable-ratio reinforcement schedule. Water delivered at the unvisited goal remained available until the rat's next visit without additional water delivery. This implies that reward probability for a given target increases with the number of consecutive choices for the other target during the DAWH task. Mean (± SD) trial duration was 17.64 ± 13.35 s in the TAB task and 16.25 ± 14.82 s in the DAWH task.

Monkeys performed an intertemporal choice task (*Cai et al., 2011*; *Kim et al., 2008*). A trial began with the monkey's fixation of gaze on a white square presented at the center of a computer screen. Following a 1 s fore-period, two peripheral targets were presented. One target was green and delivered a small reward (0.26 ml of apple juice) when it was chosen, whereas the other target

was red and delivered a large reward (0.4 ml of apple juice). The number of yellow disks (n = 0, 2, 4, 6, or 8) around each target indicated the delay (1 s/disk) between the animal's choice and reward delivery (0 or 2 s for a small reward; 0, 2, 4, 6, or 8 s for a large reward). Each of the 10-possible delay pairs for the two targets was displayed four times in alternating blocks of 40 trials in a pseudo-random manner with the position of the large-reward target counterbalanced.

## Reinforcement learning models

We used the Q-learning model (*Sutton and Barto, 1998*) to calculate the action values ($Q_L$ and $Q_R$ for left-target and right-target choices, respectively) for the TAB task, and the stacked-probability model (*Huh et al., 2009*) for the DAWH task, respectively. In the Q-learning model, action values ($Q_a(t)$) were computed in each trial as follows:

$$\text{if } a = a(t), \ Q_a(t+1) = (1-\alpha)Q_a(t) + \alpha R(t)$$
$$\text{else } Q_a(t+1) = Q_a(t), \tag{2}$$

where $\alpha$ is the learning rate, $R(t)$ denotes the reward in the $t$-th trial (1 if rewarded and 0 otherwise), and $a$ indicates the selected action (left or right goal choice). In the stacked-probability model, values were computed considering that reward probability of the unchosen target increases as a function of the number of consecutive alternative choices (see *Huh et al., 2009* for details).

For the intertemporal choice task (*Cai et al., 2011*; *Kim et al., 2008*), the temporally DV was computed using a hyperbolic discount function as the following:

$$DV_x = A_x / (1 + kD_x), \tag{3}$$

where $A_x$ and $D_x$ indicate the magnitude and the delay of the reward from target $x$, and the parameter $k$ determines the steepness of the discount function. We indicate action value as $DV_x$ instead of $Q_a$ to denote temporally DV in the monkey studies. Actions were chosen according to the softmax action selection rule in all models as the following:

$$P_L(t) = \frac{1}{1 + \exp(-\beta(Q_L - Q_R) + b)}, \tag{4}$$

where $P_L(t)$ is the probability to choose the left goal, $\beta$ is the inverse temperature that defines the degree of randomness in action selection, $b$ is a bias term for selecting the left target, and $Q_L$ and $Q_R$ (or $DV_L$ and $DV_R$) are values associated with two alternative actions of choosing left and right targets, respectively, in trial $t$. All the model parameters were estimated using a maximum likelihood method.

## Regression analysis

We used multiple linear regression models to identify neurons related to action value or chosen value. For action-value-related neural activity, we analyzed neural spikes during the delay period (before action selection) using several different regression models. The simplest contained only the left and right action values as explanatory variables as follows:

$$S(t) = a_0 + a_1 Q_L(t) + a_2 Q_R(t) + \varepsilon(t), \text{ (model 1)} \tag{5}$$

where $S(t)$ is the spike count in a given analysis time window in trial t, $Q_L(t)$ and $Q_R(t)$ are the action values for the left and right target choices, respectively, and $\varepsilon(t)$ is the error. The majority of the analysis was based on the following model that contained the animal's choice (*C*, 1 if left and 0 if right) and chosen value ($Q_c$) as additional explanatory variables to control for effects of these variables on action values:

$$S(t) = a_0 + a_1 Q_L(t) + a_2 Q_R(t) + a_3 Q_c(t) + a_4 C(t) + \varepsilon(t), \text{ (model 2)} \tag{6}$$

We subjected this model to various resampling-based tests to identify action-value neurons. To compare the results from our previous analysis method (*Sul et al., 2010*; *Sul et al., 2011*; *Lee et al., 2012b*; *Kim et al., 2013*; *Lee et al., 2017*), we added AR terms, namely neural spikes during the same analysis time window in the previous three trials, to model 2.

$$\mathrm{AR} = a_5 S(t-1) + a_6 S(t-2) + a_7 S(t-3),$$

To investigate how multiple types of value signals are encoded in the activity of neurons across different brain areas, we tested a regression model that includes the sum of action values, $\Sigma Q(t) = Q_L(t) + Q_R(t)$, and their difference, $\Delta Q(t) = Q_L(t) - Q_R(t)$, which roughly correspond to state value and policy, respectively.

$$S(t) = a_0 + a_1 \sum Q(t) + a_2 \Delta Q(t) + a_3 Q_c(t) + a_4 C(t) + \varepsilon(t), \ (\mathrm{model}\ 3) \tag{7}$$

This regression model would fit the data equally well compared to the model containing action values ($Q_L$ and $Q_R$) because $\Delta Q$ and $\Sigma Q$ are linear combinations of action values. For chosen-value-related neural activity recorded in rats at the time choice outcome was revealed, the following two regression models were used:

$$S(t) = a_0 + a_1 Q_c(t) + \varepsilon(t), \ (\mathrm{model}\ 4) \tag{8}$$

$$S(t) = a_0 + a_1 Q_L(t) + a_2 Q_R(t) + a_3 Q_c(t) + a_4 C(t) + a_5 R(t) + a_6 X(t) + \varepsilon(t), \ (\mathrm{model}\ 5) \tag{9}$$

where $R(t)$ is reward (1 if reward and 0 if unrewarded) and $X(t)$ is the interaction between choice and reward.

Action-value-related neural activity in the monkey was analyzed using the following regression model:

$$S(t) = a_0 + a_1 DV_L(t) + a_2 DV_R(t) + a_3 (DV_{chosen}(t) - DV_{unchosen}(t)) + a_4 C(t) + \varepsilon(t), \ (\mathrm{model}\ 6) \tag{10}$$

where $DV_L(t)$ and $DV_R(t)$ are temporally DVs for the left and right target choices, respectively, and $DV_{chosen}(t)$ and $DV_{unchosen}(t)$ are temporally DVs for the chosen and unchosen target choices, respectively. Neural activity related to the sum of and difference between temporally DVs ($\sum DV$ and $\Delta DV$, respectively) was assessed with the following regression model:

$$S(t) = a_0 + a_1 \sum DV(t) + a_2 \Delta DV(t) + a_3 (DV_{chosen}(t) - DV_{unchosen}(t)) + a_4 C(t) + \varepsilon(t), \ (\mathrm{model}\ 7) \tag{11}$$

## Permutation and surrogate data-based tests

For the session permutation and pseudosession tests, value-related neural activity was assessed using spike data of the original session. In the session permutation test, the original neural data was paired with 382 remaining behavioral sessions. The results did not differ qualitatively when we paired the neural data only with the same type of behavioral sessions as the original one (214 TAB and 167 DAWH remaining sessions). In the pseudosession test, we generated 500 simulated behavioral sessions based on the Q-learning (for TAB-task sessions) or stack probability (for DAWH-task sessions) model using model parameters estimated for a given animal. For the FPR and AAFT tests (*Theiler et al., 1992*), value-related neural activity was assessed using the original behavioral data and 1000 samples of surrogate neural data. In the FPR test, each surrogate neural data was generated with the same amplitude of the Fourier transform as the original data but with random phase. In the AAFT test, the same number of elements as the number of trials in the original neural data was drawn randomly from a Gaussian distribution, and these elements were then sorted according to the rank of the neural data (Gaussianization). All zero values (no spikes) of the neural data were replaced with small (<1) randomly chosen nonzero values in order to avoid artifacts in sorting consecutive zero values. The FPR method was then applied to the sorted Gaussian data. Finally, the original neural data was reordered according to the rank of the phase-randomized Gaussian data (de-Gaussianization), and this reordered neural data was used as surrogate neural data. For comparison, we also tested the within-block permutation procedure we used in our previous study (*Kim et al., 2009*). For this, we randomly shuffled spike data 1000 times across different trials within each block while preserving the original block sequence.

## Statistical analysis

Significance (p-value) of a regression coefficient was determined with the *t*-test or by the frequency in which the absolute magnitude of *t*-value for the regression coefficient obtained using a

permutation test or surrogate data exceeds that of the original *t*-value (resampling-based tests). Statistical significance of the fraction of action-value or chosen-value neurons in a given brain area was determined based on the binomial test.

To examine how different types of value signals are represented across different brain areas, we exploited the fact that the neurons encoding specific types of value signals, such as action values or policy, would be distributed along an oriented line through the origin in a complex plane defined by $z = R \cdot e^{-i\theta} = a_L + a_R \cdot i$, where $R = \sqrt{a_L^2 + a_R^2}$, $\theta = \text{atan}(a_R/a_L)$, $i = \sqrt{-1}$, and $a_L$ and $a_R$ are regression coefficients for left and right action values, respectively (namely, $a_1$ and $a_2$ in *Equationd 6 and 10*). In this plane, neurons encoding state value or policy would display twofold rotational symmetry, since they would be distributed mainly along the lines defined by $y = x$ or $y = -x$. When the angles are doubled, $Q_L$- and $Q_R$-coding neurons would form vectors along the x-axis ($Q_L$, positive; $Q_R$, negative) while $\Sigma Q$- and $\Delta Q$-coding neurons would form vectors along the y-axis ($\Sigma Q$, positive; $\Delta Q$, negative). Therefore, we examined the vertical component of the mean resultant vector after multiplying the angle of the vector z by a factor of 2 in order to test whether neurons in a given area tended to encode policy or state value more strongly. By contrast, the neurons encoding action values would show fourfold rotational symmetry since they would be clustered around $x = 0$ or $y = 0$. Therefore, we examined the horizontal component of the mean resultant vector after multiplying the angle of z by a factor of 4 in order to test whether neurons tended to encode action values of individual choices or combine them for policy or state value. We used Wilcoxon rank-sum test to determine whether the horizontal or vertical component of the mean vector was significantly different from 0, and one-way ANOVA and Bonferroni post hoc tests to test whether they significantly varied across regions.

Throughout the paper, p=0.05 was used as the criterion for a significant statistical difference unless noted otherwise. Data are expressed as mean ± SEM unless noted otherwise. Raw data of this work is archived at Dryad (https://doi.org/10.5061/dryad.gtht76hj0).

## Acknowledgements

This work was supported by the Research Center Program of Institute for Basic Science (IBS-R002-A1; MWJ) and the National Institute of Health grants (DA 029330; DL).

## Additional information

### Competing interests

Daeyeol Lee: Reviewing editor, *eLife*. The other authors declare that no competing interests exist.

### Funding

| Funder | Grant reference number | Author |
|---|---|---|
| Institute for Basic Science | IBS-R002-A1 | Min Whan Jung |
| National Institute of Mental Health | DA 029330 | Daeyeol Lee |

The funders had no role in study design, data collection and interpretation, or the decision to submit the work for publication.

### Author contributions

Eun Ju Shin, Yunsil Jang, Conceptualization, Data curation, Formal analysis, Validation, Investigation; Soyoun Kim, Data curation, Formal analysis; Hoseok Kim, Xinying Cai, Hyunjung Lee, Jung Hoon Sul, Sung-Hyun Lee, Data curation; Yeonseung Chung, Formal analysis, Supervision; Daeyeol Lee, Min Whan Jung, Conceptualization, Supervision, Validation

## Author ORCIDs

Eun Ju Shin (iD) https://orcid.org/0000-0003-3901-498X
Yunsil Jang (iD) https://orcid.org/0000-0001-5736-214X
Soyoun Kim (iD) http://orcid.org/0000-0003-1348-6401
Daeyeol Lee (iD) https://orcid.org/0000-0003-3474-019X
Min Whan Jung (iD) https://orcid.org/0000-0002-4145-600X

## Decision letter and Author response

Decision letter https://doi.org/10.7554/eLife.53045.sa1
Author response https://doi.org/10.7554/eLife.53045.sa2

---

# Additional files

## Supplementary files

• Supplementary file 1. Statistical test results for 2θ and 4θ plots. Top, statistical test results for 2θ plots. Orange shading, Y-component of the mean vector was tested for significant deviation from 0 (Wilcoxon rank-sum test, red indicates p-values <0.05). No shading, Y-component of the mean vector was compared across regions using one-way ANOVA (rat, $F_{(7,2587)} = 12.64$, $p=4.9 \times 10^{-16}$; monkey, $F_{(2,247)} = 10.75$, $p=3.4 \times 10^{-5}$) followed by Bonferroni post hoc tests. Significant differences (p-values <0.05) between regions are indicated in red. Bottom, statistical test results for 4θ plots. Orange shading, X-component of the mean vector was tested for significant deviation from 0 (Wilcoxon rank-sum test, red indicates p-values<0.05). No shading, X-component of the mean vector was compared across regions using one-way ANOVA (rat, $F_{(7,2587)} = 3.79$, $p=4.3 \times 10^{-4}$; monkey, $F_{(2,247)} = 0.95$, $p=0.387$) followed by Bonferroni post hoc tests. Significant differences (p-values <0.05) between regions are indicated in red.

• Transparent reporting form

## Data availability

All data generated or analyzed during this study are included in the manuscript and supporting files. Raw data to reproduce this work is archived at Dryad https://doi.org/10.5061/dryad.gtht76hj0.

The following dataset was generated:

| Author(s) | Year | Dataset title | Dataset URL | Database and Identifier |
|---|---|---|---|---|
| Shin EJ, Jang Y, Kim S, Kim H, Cai X, Lee H, Sul JH, Chung Y, Lee D, Jung MW | 2021 | Data from: Robust and distributed neural representation of action values | https://doi.org/10.5061/dryad.gtht76hj0 | Dryad Digital Repository, 10.5061/dryad.gtht76hj0 |

---

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
