## [Decision Letter]

**Acceptance summary:**

Action values are important components in reinforcement learning. Single neutrons in the brain have been reported to signal these values, but recent work has suggested that problems with these analyses bring these data into question. This paper performs rigorous analysis to show the action value signals are robust. In doing so it contributes to an important line of technical research that is finding safer ways to analyse neuronal data.

**Decision letter after peer review:**

Thank you for submitting your article "Further evidence for neural representation of action value" for consideration by *eLife*. Your article has been reviewed by 3 peer reviewers, and the evaluation has been overseen by Timothy Behrens as the Senior Editor and Reviewing Editor. The reviewers have opted to remain anonymous.

The reviewers have discussed the reviews with one another and the Reviewing Editor has drafted this decision to help you prepare a revised submission.

Summary:

The manuscript is a response to recent work (Elber-Dorozko and Loewenstein 2018) showing that inferences about action value coding in neural population can be distorted by two mechanisms: (1) Serial correlation from trial to trial in both neural activity and action values, that causes statistical analyses that assume independence of each trial to overestimate significance (2) Correlation between action values and other behavioural/decision variables which can cause incorrect inference that neurons coding for other variables represent action values.

The present study uses simulations and reanalysis of neuronal recordings to address methodological and scientific questions raised by Elber-Dorozko and Loewenstein's work. Broadly, they do so convincingly (although not elegantly) with respect to serial correlations. The portion of the manuscript that deals with correlated variables is less convincing, but this is an issue of narrower significance.

The manuscript is of interest to *eLife* for the following reasons. The previous paper raised an important methodological concern that had been uniformally ignored in the analysis of neuronal activity across many fields. In doing so, however, it also cast doubts on previous results that had studied one particular computation performed by neuronal activity – the computation of action value. These were important studies revealing a fundamental computation underlying how the brian controls behaviour. The current manuscript acknowledges the methodological concern but allays the doubts over action-value computations. It is therefore of potential significant interest.

Essential revisions:

The methods used to investigate and account for serial autocorrelations are coarse, ad-hoc, and little is presented to verify that they do what they say they do. This raises 2 issues. The first relates to how robust the findings will be in the face of future related criticisms. The second relates to setting methodological standards for how the field should deal with serial correlations in the future.

We will deal with the second issue first, as dealing with it properly may make the first problem redundant.

Strategy for accounting for autocorrelations:

The issue is that the authors propose 3 techniques for dealing with serial autocorrelations in the noise (simulating neurons with serial correlations, permuting "within blocks" only, and including a few trials back as co-regressors). None of them are elegant or general techniques, and in all cases the behaviour of these techniques in the null case is poorly characterised (see below for more on this point). This is particularly surprising because this is an old problem in the statistics literature, and there are off-the-shelf techniques for addressing the problem elegantly and rigorously. We acknowledge the requirement to adhere to the spirit of the EDL paper, but we think it would be extremely advantageous to move the argument forward using general well-validated techniques.

Options include

1. Estimating a whitening kernel for the residuals and refitting the model using this kernel (technically nontrivial). For an example from fMRI, see Woolrich et al. Neuroimage 2001.

2. Fitting autoregressive noise models using standard off-the-shelf software (eg gls in r – see review comment below ).

3. Permutation tests after transformations that render the data exchangeable.

Eg fourier transform the data and permute only the phases then reconstruct the data.

Eg Wavelet transform the data and permute wavelet coefficients then reconstruct the data.

Here are the reviewer comments that led to this discussion, which contain other related comments that maybe useful. A similar discussion was had at triage. We note that, whilst the reviewer suggests an auto-regressive model, which would be fine, it would also be fine to use one of the other techniques above, which may be more appropriate if the residuals are not well described by a limited AR model such as AR(1). The permutation strategies described above are trivial to implement and effective.

"I think the paper could do a better job technically unpacking the issues with temporal correlations, which in my view weren't diagnosed as precisely as they could have been in the original article either. The senior author knows more econometrics than I do, but as I understand it, all of the estimation issues in OLS here are due to the assumption of uncorrelated *errors*. Autocorrelation in the dependent variables, and the explanatory variables, is perfectly OK (indeed, one would result from the other) so long as they cancel each other out when the model is fit, in which case the residuals would be uncorrelated. Changing the autocorrelation in either y or X affects this only indirectly, since both appear in the residual. I think the article's focus on the autocorrelation in the explanatory variables and spike rates – both in rhetoric and analysis and results – is a piece of the puzzle but tends to obscure this deeper point. It would be helpful to also focus on visualizing and decorrelating the residuals. For the same reason, the lack of *regressor* autocorrelation in the monkey experiment is less of a solution than it is made out to be, I think."

"I find the estimation strategy of Figure 3 and sporadically onward a bit frustrating and less convincing (or at least more roundabout) than it could be. I'm sympathetic to the overall conclusion, but the overall strategy comes off as piling up multiple fixes, even though they are shown not to work adequately on simulated data. To compensate for this, the simulations are used to define a new, inflated false positive rate that is, finally, in a followup test, compared to the obtained rate of nominal positives. Frankly: yuck. How about figuring out why the fixes don't work, and finding a test that does work? For the nonparametric fix (bootstrap) the issue is presumably within session correlations, as discussed later; but for the lagged AR terms, I assume the problem is there aren't enough of them to handle longer-timescale correlations. But this is itself kind of a hack; a more orthodox parametric approach would be to use a nonlinear, generalized least squares (eg gls() in R) to estimate a full AR(1) model or whatever other error covariance form is supported by the actual data. (Note that even an AR(1) process predicts correlations at arbitrary lags so adding individual lag terms is not sufficient.)"

Characterisation of performance in the null case.

If the authors change their strategy as recommended above, this section of the review may be rendered redundant. However, given the current approach the review team did not think that the paper did a good job in presenting diagnostics that adequately evaluate the performance of their strategies.

Minimally, given how much work the random walk neuron model does, we think that the authors should try harder to evaluate the performance with a model that looks more like the data. The model was setup only to match the neuronal autocorrelations at lag 1 trial and likely has a very different autocorrelation structure from real neurons at lags greater than 1. The autocorrelation structure of the control 'random' neuron model should be matched to that of the neuronal data. This may need a generative model that is more expressive than AR(1). Without this, the authors are susceptible to future criticism that simply shows that the authors techniques do not do well in the face of realistic data.

We also think that instead of simply reporting the number of false positives at p<0.05 threshold, the authors should construct the p-p plot (Wikipedia), which plots observed false positives in empirical data against the nominal threshold. This will make it useful to future researchers who would like to use the same techniques with different threshold.

All three reviewers made the same point. I include all 3 here to encourage the authors that it is an important point that will likely be shared by many readers.

"Generation of random-walk neurons. How is it possible to create the same autocorrelation kernel as the one observed in the neural data (essentially flat – at least for the shown scale of 5 trials) through a random-walk process – for which the correlation should intrinsically decrease over time? The authors mentioned that they have matched autocorrelation at lag 1 only, which may be good enough as an approximation for what the authors intend to do with random-walk neurons, but it is not a tight match and the authors may want to mention this somewhere in the manuscript."

"The random walk neuron model does a lot of work as a control against which real neurons are compared. However, the model was setup only to match the neuronal autocorrelations at lag 1 trial and likely has a very different autocorrelation structure from real neurons at lags greater than 1. The autocorrelation structure of the control 'random' neuron model should be matched to that of the neuronal data. "

"Either way, everything comes down here to the simulated spike trains under the null model, and it would be good to have more argument that these are actually a good simulation for the data. Among other things, I wasn't clear if their timescale is individually fit per brain area or experiment or just roughly chosen; if multiple timescales of correlation are detectable in the actual data, rather than just rectified AR(1) as here; and again if the autocorrelative structure of the residuals is similar between data and simulation. "

Action values vs policy etc.

There was broad scepticism amongst the reviewers as to whether it was possible to dissociate policies from values, and whether it was really relevant to do so, particularly if policy is (confusingly) used to refer to a difference in Q-values. This is reflected in the comments below. Whilst we acknowledge the authors' ambitions to address the critiques raised in EDL, we encourage great care in the interpretation of this whole section. Again, related points were made by all 3 reviewers, highlighting that this Is likely also to be an issue for many readers.

"I find the second half of the article, on alternative decision variables, a little bit of a red herring. One thing is that the relationship between a Q value and a policy (as the term is normally used in RL, and was used by Elber-Doroko) is nonlinear. Calling the difference in Q values a "policy" is just not using the term accurately. On the other hand, my view is that this example shows that the whole critique is ill founded, and the only useful question is what is the (linear or nonlinear) relationship between decision variables and brain activity. Neural representations of values are likely to be nonlinear for reasons other than policy (eg, there is plenty of work by Glimcher and others on gain control or divisive normalization) and may also be differential (eg, activity which is related to the relative value, chosen minus unchosen, which is nevertheless in units of value and not normalized/softmaxed etc into a policy). Telling the difference between divisive and subtractive normalization is not really viable, especially in the linear setting; and even so, the same (softmax) algebraic form could describe either policy or (gain controlled) value. There's just not a meaningful categorical distinction to be made. I suppose there might be some way of recasting this section to focus on the distinction between summation vs difference as being representative (in a linear framework) of state values vs. relativized, or normalized, or post-choice policy values. But I think it's giving too much away to frame this as actually distinct variables confounding one another; and also unfair to call a difference a policy."

"Correlation with sum(Q) and diff(Q). I don't understand the exact graphical description on Figure 8 and Figure 8. The authors label gray neurons as 'only Q', but many of them are probably not coding anything (non-significant, corresponding to black neurons in Figure 6 of the article by EDL). Also, I expected that the neurons coding selectively for one action value (QL or QR) should be found on Figure 8A for |x| > threshold and |y| ~ 0 and vice versa. However, it is clearly not the case given the labelling of neurons provided by the authors for this graph. Could the authors clarify this and explain the apparent discrepancy with the analyses performed by EDL (Figure 6 from their article). I have a similar concern regarding Figure 8B: pure action-value neurons seem to be located only at the center of the graphs (for |x| ~ 0 and |y| ~ 0), which is where non-selective neurons should be found."

"Figures 7 – 9 attempt to dissociate action value coding from coding of policy (difference in action values) and state value (approximated as sum of action values). As these variables are linearly dependent, it is formally impossible to say whether a neuron represents one of them, or a linear combination of the others. Mixed linear selectivity is ubiquitous (e.g. Kobak et al. *eLife* 2016;5:e10989), so it not that interesting to ask which of this degenerate set of variables is most 'purely' represented by each neuron. This said, having chosen a given non-degenerate pair of these variables to work with, it is interesting to know how representation of one variable correlates with representation of the other across the population, and how this varies across regions. This is shown nicely in figure 7A and the top two panels of 9A, but I felt the remaining panels of figures 7-9 did not add additional value."

When the authors assess the extent of chosen-value coding (Figure 5, 6B) , they include the individual action values in their regression model, which is important as these variables are correlated. However, when they assess action value coding (Figures 4, 6A) they do not include chosen value in the models. I think the rationale is that the analyses are different trial epochs, pre-choice for 4, 6A, post outcome for 5, 6B. However, chosen-value coding is certainly possible before the choice is executed and hence chosen value should be included in the model when assessing action value coding.

[Editors' note: further revisions were suggested prior to acceptance, as described below.]

Thank you for submitting your article "Further evidence for neural representation of action value" for consideration by *eLife*. Your revised article has been reviewed by 3 peer reviewers, and the evaluation has been overseen by Timothy Behrens as the Senior Editor and Reviewing Editor. The reviewers have opted to remain anonymous.

I have written a summary of our opinion after the discussion directly below here. I have also left the reviews below for emphasis and detail, but please don't feel that you need to address all the points in the reviews. If you can address the central issues in the summary directly below here, we will be happy.

This revision has raised some complications in the reviewers' minds that have led to a lot of discussion. In brief, we are not happy with the 2-stage approach that does not lead to p-values for individual neurons that can be trusted.

Whilst we agree that this approach goes some way to rebutting the EDL finding, it is a narrow rebuttal which does not provide a good way forward for scientists faced with similar problems in the future. The combination of GLS modelling that does not accurately deal with the autocorrelations, with non-standard application of circular permutations to demonstrate control performance is, in our view, dangerous, and not useful to the community. We are not keen on publishing such an approach in *eLife*.

If the GLS approach does not lead to good corrections for autocorrelations, then we think it incumbent upon you guys to find a non-parametric approach that does.

We are slightly bemused by your assertion that you cannot do the Fourier permutation test because the timepoints are not equally sampled. There are two reasons we are bemused. The first is that the same argument applies to the circular permutation that you do use. The second is that there are well-established methods for computing Fourier transforms for non-uniformly sampled data. However, we think that you don't even need to use them. We are happy for you to line the trials up in a matrix and do the Fourier transform across trials. We think the danger of introducing errors by this approximation is small.

It is also possible that a correct application of the circular method you propose would work (i.e. ignore GLS and just do the full circular permutation test). The three likely issues with this are (1) the circular permutations might correlate with the original design matrix leading to low sensitivity, (2) edge effects will mean that the circular-shifted test will have different autocorrelation properties than the original test. This can lead to false positives (and actually may be a problem in your control analysis in the paper). (3) The small number of possible permutations will prevent accurate inference in the tail (and possibly prevent eg corrections for multiple comparisons).

Note that although reviewer 2 below has an issue with the >10 trial threshold for the circular test, this was not thought a problem after discussion.

In order to assess these effects, one of the reviewers prepared a Jupyter Notebook comparing the different approaches. It looks as though I can only attach a single file to the letter and the notebook comes in both.html and.py forms. I have attached the.html. If you email me when you get this, I will forward the (anonymised).py.

You can see that, whilst both Fourier and Circular approaches suffer slightly from edge effects, the circular test is much more severely affected. There are readily available techniques in the literature for removing edge effects. For example you could window the data before the regression (eg using a Tukey window). We think it would likely be profitable for you to investigate these approaches whichever permutation method you choose. You can see, however, that even without dealing with these details, the Fourier method does a pretty good job.

You can see in the reviews below, that the reviewers also remain concerned about the distinction in the second half of the paper between action values, chosen values, policies etc, which interacts heavily with questions about the linearity of neuronal responses. We remain concerned that there is a danger this will confuse more than clarify the issue for the community. However, we realise that there is a similar section in the EDL paper, and that you guys need to address this. This may be TB's fault (along with the original reviewers) for not flagging this in the original paper. We would appreciate a clear statement in this section in the paper that states that this section is a narrow rebuttal of the EDL paper, and discusses the difficulties in differentiating policy from action values etc. (see reviews below).

*Reviewer #1:*

The authors have substantially modified their manuscript, including the general statistical approach for assessing the neural encoding of value signals. Doing so, they have addressed several of the concerns I have regarding the original manuscript. The GLS regression approach appears to support the main claims made in the original manuscript. Although I found at times the description of the results (including their illustration) to be less clear than before, I do not have important concerns that remain to be addressed.

*Reviewer #2:*

The authors have taken some welcome steps to address reviewer comments. They now use a regression model which explicitly models autocorrelation in the residuals (GLS model), they compare models with different order auto-regressive structure using BIC, and show P-P plots for real and circularly permuted data. These steps do improve the manuscript, but unfortunately the statistical approach still has real problems.

The main request of the reviews was: 1. Use a principled method which would fix the problem of P value inflation due to correlations. 2. Show that it works (i.e. gives correct P values) using P-P plots on simulated or otherwise generated 'null' data. 3. Use P values derived from this method directly as the statistics reported.

Doing this properly solves the issue once and for all, and both answers the specific question (do neurons encode action values) and provides a method the field can use to avoid problems in future.

What the authors have actually done is; 1. Use a principled method (GLS model with autocorrelated residuals) to obtain P values. 2. Provide diagnostics based on null data generated via a poorly implemented circular permutation approach (see below), which none the less suggests that the GLS model has not fixed the problem (P values for null data are still inflated). 3. Test whether the fraction of significant neurons in the original data is significantly different from the average fraction of significant neurons across the permuted data.

This is really not good, as it does not provide a way of calculating accurate P values for individual neurons, nor does it take into account variability across permutations when asking if the fraction of significant neurons in the real data is significantly higher than that expected by chance.

Circular permutation of neuronal data relative to behavioral data by a random number of trials is a good way of generating data under the null hypothesis that there is no relationship between activity and behavior, while preserving the autocorrelations in both. However, the authors did their permutations "with the constraint that the minimum difference of trial number between the original and shifted data is > 10". This makes the permuted data meaningless as it no longer comes from any well-defined null distribution. Additionally, to accurately estimate the distribution of the measure of interest under the null hypothesis, a large number of permutations is needed (thousands), while the current work used only 10. As the authors only used the mean across permutations rather than the distribution when calculating their statistics, the number of permutations is perhaps less problematic, but this is a very non-standard way to use permutations.

In my understanding, the correct permutation test to generate accurate P values in a regression analysis of neuronal activity is as follows: 1. Calculate the measure of interest on the real data. This could be a β weight for a particular neuron, or a summary measure across the population (e.g. average β squared or coefficient of partial determination across neurons). Summary measures across the population will have much more statistical power once you have more than a few neurons. 2. Generate an ensemble of e.g. 5000 permutated datasets. To make each permuted dataset, circularly permute the neuronal data relative to the behavioral data by a random number of trials between 1 and the number of trials in the session. If multiple sessions contribute to your measure of interest, draw the circular shift separately for each session for each permuted dataset. 3. Calculate the measure of interest for each permuted datasets. The distribution of the measure across permuted datasets is an estimate of it's distribution under the null hypothesis that there is no relationship between behavior and neural activity. Calculate a P value by comparing the value of the measure for the real data with its distribution across the permutations. For a two tailed test the P value is min(X, 1-X) where X if the fraction of permutations for which the measure on true data is greater than that on permuted data. By construction, the P values generated by this method for circularly permuted data are uniformly distributed between 0 and 1.

*Reviewer #3:*

This article is improved but many of the core problems we identified in the original are unchanged, and overall I still feel it has promise but is not yet in publishable shape.

On the two sets of results separately:

Serial autocorrelation: The rhetoric is much more precise and improved, and I appreciate the move toward GLS and toward de-emphasizing the problematic 'null' simulations (which, though not improved are probably sufficient for the specific use to which they are now put).

However, I just think the methods presented here still haven't convincingly solved the problem at hand. The bottom line (from what I can see) is they still don't have a test for significance that actually produces correct p values, as shown by all the p/p plots. In particular, GLS with the chosen AR structure also produces inflated false positives, when run on the shifted null control data. (That said, it is hard to rule out that the problem might be due at least in part to the shifted null being overconservative for the same reason session-level permutations are, i.e. the block structure of conditions.)

The article thus continues to resort to the two-stage procedure of testing significance, criticized previously: using a demonstrably flawed method, then testing whether the proportion of significant neurons exceeds a measure of the proportion expected due to inflation. This is arguably valid, for the narrow job of rejecting the claim that previous results, in the aggregate, are due to p inflation, but it simply isn't a viable procedure going forward for conducting inference neuron by neuron. Just for instance, the very next section of the paper (on policy and value coding) contains extensive discussion counting and comparing the number of nominally significant neurons of different types, but none of these numbers can be taken seriously given what immediately preceded.

My view is that for the work to be useful to the field, it needs to present a method that gives a demonstrably trustworthy p value at the single neuron level. If GLS doesn't work, and they must resort to augmenting it with a further nonparametric stage, then they may as well just instead go ahead and define a proper nonparametric test, e.g. a full permutation test based on the circular shift. In this case the GLS is moot and correlation would work fine as the test statistic. The main question to my eye here is the validity / independence / sensitivity of the circular shifts as the unit of permutation. Given very long autocorrelation, many such shifts will be non-independent from each other, and furthermore I don't completely understand why same the issue argued to plague the across-session permutation control (i.e. nonindependence due to structure in the trial blocking) doesn't also apply here. Thus, I think to go this way would ideally require more work to validate the control, but I fear this means they are back with the problem of designing convincing null simulations.

Value vs policy coding: I continue to think this chunk of the paper is mostly built on confusing conceptual foundations, admittedly mostly inherited from the earlier paper. First, I still think the whole motivating framing that activity related to action value isn't bona fide action value activity if it is negatively modulated by the alternative value ("policy") is just plain wrong, for many reasons I rehearsed before: related for instance to normalization and efficient coding. Second, this linear approach neglects chosen value altogether (but is confounded by it), which other results in the paper suggest are a key factor.

Finally, apart from the fact that the p values themselves are dubious (see above) the exercise of counting neurons that are significant or nonsignificant on different sets of correlated tests does not clearly lend itself to any formal conclusion or even informal interpretation. What results would be expected under different hypotheses? What are the hypotheses? The idea of "mixed selectivity" is neither defined nor tested, and I don't see that this would be a viable way to do it: ultimately, interpreting the Venn diagram of positive and negative results on correlated tests flirts with a combination of the fallacies of frequentist reasoning including affirming nulls and double dipping. The analysis of angles is much less problematic in this respect (since it represents a single test with a well-defined null hypothesis rather than a family of non-independent tests), though again it is a bit less than one might hope for since it is at the population, rather than neuron level. I'd still vote to axe this section entirely, or narrow it way down to the angle thing if necessary.

---

## [Author Response]

Essential revisions:The methods used to investigate and account for serial autocorrelations are coarse, ad-hoc, and little is presented to verify that they do what they say they do. This raises 2 issues. The first relates to how robust the findings will be in the face of future related criticisms. The second relates to setting methodological standards for how the field should deal with serial correlations in the future.

Two major concerns raised by the reviewers were (1) the lack of systematic approach to evaluate statistical significance of value-signals in neural activity in the presence of residual autocorrelation, and (2) conflation of signals related to the contrast of action values and policy. To address these concerns, we have performed an almost completely new series of analyses on the neural data and re-wrote much of the entire manuscript. In particular, based on the suggestions from the reviewers and advice from an expert on applied statistics (a new co-author in the manuscript), we have adopted the generalized least square (GLS) regression model to evaluate the nature of residual autocorrelation in neural activity, and dramatically simplified the permutation tests to evaluate the statistical significance of value signals in neural activity. We also added the results obtained from the monkey dorsolateral prefrontal cortex. These and other major changes in the revised manuscript are summarized below.

We will deal with the second issue first, as dealing with it properly may make the first problem redundant.Strategy for accounting for autocorrelations:The issue is that the authors propose 3 techniques for dealing with serial autocorrelations in the noise (simulating neurons with serial correlations, permuting "within blocks" only, and including a few trials back as co-regressors). None of them are elegant or general techniques, and in all cases the behaviour of these techniques in the null case is poorly characterised (see below for more on this point). This is particularly surprising because this is an old problem in the statistics literature, and there are off-the-shelf techniques for addressing the problem elegantly and rigorously. We acknowledge the requirement to adhere to the spirit of the EDL paper, but we think it would be extremely advantageous to move the argument forward using general well-validated techniques.Options include1. Estimating a whitening kernel for the residuals and refitting the model using this kernel (technically nontrivial). For an example from fMRI, see Woolrich et al. Neuroimage 2001.2. Fitting autoregressive noise models using standard off-the-shelf software (eg gls in r – see review comment below ).3. Permutation tests after transformations that render the data exchangeable.Eg fourier transform the data and permute only the phases then reconstruct the data.Eg Wavelet transform the data and permute wavelet coefficients then reconstruct the data.Here are the reviewer comments that led to this discussion, which contain other related comments that maybe useful. A similar discussion was had at triage. We note that, whilst the reviewer suggests an auto-regressive model, which would be fine, it would also be fine to use one of the other techniques above, which may be more appropriate if the residuals are not well described by a limited AR model such as AR(1). The permutation strategies described above are trivial to implement and effective."I think the paper could do a better job technically unpacking the issues with temporal correlations, which in my view weren't diagnosed as precisely as they could have been in the original article either. The senior author knows more econometrics than I do, but as I understand it, all of the estimation issues in OLS here are due to the assumption of uncorrelated errors. Autocorrelation in the dependent variables, and the explanatory variables, is perfectly OK (indeed, one would result from the other) so long as they cancel each other out when the model is fit, in which case the residuals would be uncorrelated. Changing the autocorrelation in either y or X affects this only indirectly, since both appear in the residual. I think the article's focus on the autocorrelation in the explanatory variables and spike rates – both in rhetoric and analysis and results – is a piece of the puzzle but tends to obscure this deeper point. It would be helpful to also focus on visualizing and decorrelating the residuals. For the same reason, the lack of regressor autocorrelation in the monkey experiment is less of a solution than it is made out to be, I think.""I find the estimation strategy of Figure 3 and sporadically onward a bit frustrating and less convincing (or at least more roundabout) than it could be. I'm sympathetic to the overall conclusion, but the overall strategy comes off as piling up multiple fixes, even though they are shown not to work adequately on simulated data. To compensate for this, the simulations are used to define a new, inflated false positive rate that is, finally, in a followup test, compared to the obtained rate of nominal positives. Frankly: yuck. How about figuring out why the fixes don't work, and finding a test that does work? For the nonparametric fix (bootstrap) the issue is presumably within session correlations, as discussed later; but for the lagged AR terms, I assume the problem is there aren't enough of them to handle longer-timescale correlations. But this is itself kind of a hack; a more orthodox parametric approach would be to use a nonlinear, generalized least squares (eg gls() in R) to estimate a full AR(1) model or whatever other error covariance form is supported by the actual data. (Note that even an AR(1) process predicts correlations at arbitrary lags so adding individual lag terms is not sufficient.)"

We agree with the editor and reviewers that we should have dealt with the statistical issues resulting from serial correlation in neural data more systematically. We are also grateful to the reviewers for suggesting 3 specific approaches we could take, including the use of whitening kernel, generalized least square method, and appropriate transformation (e.g., wavelet). As the reviewers pointed out, some of these methods have been well established in the field of fMRI and are broadly applied. Unfortunately, it is difficult to adapt these methods to the analysis of spike data in our current study, because unlike the fMRI data (which is sampled periodically) spike counts in successive trials during our experiment are separated by highly variable inter-trial interval. Also, the number of trials in our experiments is much smaller than the number of data points analyzed in typical fMRI experiments. Accordingly, we have decided to combine the generalized least square (GLS) regression analysis models with a circular permutation test to better assess the nature of residual autocorrelation and to develop the most robust method to circumvent the difficulty in evaluating the statistical significance of value-related signals in neural activity. We agree with the sentiment expressed by the reviewers that this would be most beneficial for future studies facing the same statistical issues. For comparison with the previous study (i.e., Elber-Dorozko and Loewenstein, 2018, EDL), however, we also kept the results obtained with the OLS method and the permutation test proposed by EDL but moved them to Figure 3 supplements (Figure 3-supplement 1 and Figure 3-supplement 2). We also show residual autocorrelation in Figure 2 of the revised manuscript as suggested by the reviewers.

Characterisation of performance in the null case.If the authors change their strategy as recommended above, this section of the review may be rendered redundant. However, given the current approach the review team did not think that the paper did a good job in presenting diagnostics that adequately evaluate the performance of their strategies.Minimally, given how much work the random walk neuron model does, we think that the authors should try harder to evaluate the performance with a model that looks more like the data. The model was setup only to match the neuronal autocorrelations at lag 1 trial and likely has a very different autocorrelation structure from real neurons at lags greater than 1. The autocorrelation structure of the control 'random' neuron model should be matched to that of the neuronal data. This may need a generative model that is more expressive than AR(1). Without this, the authors are susceptible to future criticism that simply shows that the authors techniques do not do well in the face of realistic data.We also think that instead of simply reporting the number of false positives at p<0.05 threshold, the authors should construct the p-p plot (Wikipedia), which plots observed false positives in empirical data against the nominal threshold. This will make it useful to future researchers who would like to use the same techniques with different threshold.All three reviewers made the same point. I include all 3 here to encourage the authors that it is an important point that will likely be shared by many readers."Generation of random-walk neurons. How is it possible to create the same autocorrelation kernel as the one observed in the neural data (essentially flat – at least for the shown scale of 5 trials) through a random-walk process – for which the correlation should intrinsically decrease over time? The authors mentioned that they have matched autocorrelation at lag 1 only, which may be good enough as an approximation for what the authors intend to do with random-walk neurons, but it is not a tight match and the authors may want to mention this somewhere in the manuscript.""The random walk neuron model does a lot of work as a control against which real neurons are compared. However, the model was setup only to match the neuronal autocorrelations at lag 1 trial and likely has a very different autocorrelation structure from real neurons at lags greater than 1. The autocorrelation structure of the control 'random' neuron model should be matched to that of the neuronal data. ""Either way, everything comes down here to the simulated spike trains under the null model, and it would be good to have more argument that these are actually a good simulation for the data. Among other things, I wasn't clear if their timescale is individually fit per brain area or experiment or just roughly chosen; if multiple timescales of correlation are detectable in the actual data, rather than just rectified AR(1) as here; and again if the autocorrelative structure of the residuals is similar between data and simulation. "

As we mentioned above, we now use the GLS regression to better address the problems resulting from autocorrelation in the residual from the regression model. In particular, we have used the BIC to determine the order of residual autocorrelation in the GLS method. In addition, we have performed extensive analysis to develop high-order random-walk neuron models to reproduce the shape of the autocorrelation function. Unfortunately, despite extensive simulation and data analyses, we failed to develop robust models that could accurately reproduce the main features of autocorrelation in the neural data. There are at least two reasons for this difficulty. First, we came to hypothesize that autocorrelation in neural data occurs in many different time scales, which is a topic we are currently pursuing in another manuscript. Therefore, to reproduce the observed autocorrelation, we were frequently required to include a large number of autoregressive terms, which made our analysis of value signals unnecessarily complicated. Second, there is a potential gap between the “non-linear” random-walk neuron model and the framework of linear models, because the first-order autocorrelation in the latent variable (rate parameter) can potentially be disguised as displaying higher-order autocorrelation when only the outputs (counts) of such models are considered. We think these are important problems for future investigations and we are planning to pursue them, but they are clearly beyond the scope of our current study. Therefore, in the revised manuscript, we now use circular permutation and trial-shifted data as a way to generate the null data and bypass the potential issues mentioned above. We show the results obtained with AR(1) random-walk neurons only in Figure 3-supplement 2 for comparison with the results shown in EDL paper. We also show P-P plots in Figures 3-5 as suggested by the reviewers.

Action values vs policy etc.There was broad scepticism amongst the reviewers as to whether it was possible to dissociate policies from values, and whether it was really relevant to do so, particularly if policy is (confusingly) used to refer to a difference in Q-values. This is reflected in the comments below. Whilst we acknowledge the authors' ambitions to address the critiques raised in EDL, we encourage great care in the interpretation of this whole section. Again, related points were made by all 3 reviewers, highlighting that this Is likely also to be an issue for many readers."I find the second half of the article, on alternative decision variables, a little bit of a red herring. One thing is that the relationship between a Q value and a policy (as the term is normally used in RL, and was used by Elber-Doroko) is nonlinear. Calling the difference in Q values a "policy" is just not using the term accurately. On the other hand, my view is that this example shows that the whole critique is ill founded, and the only useful question is what is the (linear or nonlinear) relationship between decision variables and brain activity. Neural representations of values are likely to be nonlinear for reasons other than policy (eg, there is plenty of work by Glimcher and others on gain control or divisive normalization) and may also be differential (eg, activity which is related to the relative value, chosen minus unchosen, which is nevertheless in units of value and not normalized/softmaxed etc into a policy). Telling the difference between divisive and subtractive normalization is not really viable, especially in the linear setting; and even so, the same (softmax) algebraic form could describe either policy or (gain controlled) value. There's just not a meaningful categorical distinction to be made. I suppose there might be some way of recasting this section to focus on the distinction between summation vs difference as being representative (in a linear framework) of state values vs. relativized, or normalized, or post-choice policy values. But I think it's giving too much away to frame this as actually distinct variables confounding one another; and also unfair to call a difference a policy."

As the reviewers pointed out, the relationship between differential action values and policy is non linear. Nevertheless, in the literature (inclu ding EDL), neurons with significant effects of value difference ( ΔQ) have been frequently referred to as policy coding neurons. We have clarified the text throughout the revised manuscript to avoid unnecessary confusion on this issue.

"Correlation with sum(Q) and diff(Q). I don't understand the exact graphical description on Figure 8 and Figure 8. The authors label gray neurons as 'only Q', but many of them are probably not coding anything (non-significant, corresponding to black neurons in Figure 6 of the article by EDL). Also, I expected that the neurons coding selectively for one action value (QL or QR) should be found on Figure 8A for |x| > threshold and |y| ~ 0 and vice versa. However, it is clearly not the case given the labelling of neurons provided by the authors for this graph. Could the authors clarify this and explain the apparent discrepancy with the analyses performed by EDL (Figure 6 from their article). I have a similar concern regarding Figure 8B: pure action-value neurons seem to be located only at the center of the graphs (for |x| ~ 0 and |y| ~ 0), which is where non-selective neurons should be found.""Figures 7 – 9 attempt to dissociate action value coding from coding of policy (difference in action values) and state value (approximated as sum of action values). As these variables are linearly dependent, it is formally impossible to say whether a neuron represents one of them, or a linear combination of the others. Mixed linear selectivity is ubiquitous (e.g. Kobak et al. eLife 2016;5:e10989), so it not that interesting to ask which of this degenerate set of variables is most 'purely' represented by each neuron. This said, having chosen a given non-degenerate pair of these variables to work with, it is interesting to know how representation of one variable correlates with representation of the other across the population, and how this varies across regions. This is shown nicely in figure 7A and the top two panels of 9A, but I felt the remaining panels of figures 7-9 did not add additional value."When the authors assess the extent of chosen value coding (Figure 5, 6B) , they include the individual action values in their regression model, which is important as these variables are correlated. However, when they assess action value coding (Figures 4, 6A) they do not include chosen value in the models. I think the rationale is that the analyses are different trial epochs, pre-choice for 4, 6A, post outcome for 5, 6B. However, chosen value coding is certainly possible before the choice is executed and hence chosen value should be included in the model when assessing action value coding.

We apologize for the confusion caused by some of the figures in the original manuscript. Some of this was due to the fact that gray circles in the original manuscript represented those neurons coding only Q (Figure 7A) or DV (Figure 9) and empty squares represented those that do not encode any of these value terms, but these two symbols could not be clearly distinguished in the figures of the original manuscript. To avoid this confusion, we replaced gray circles in Figure 6 (Figure 7A and 9 in the original manuscript) with red circles. We also agree that some of the results shown in these original figures are not essential for the main conclusion of our manuscript, so the results other than the scatter plots in original Figure 7A and Figure 9 were either removed or moved to a supplementary figure (Figure 6-supplement 1). Instead, we added a new analysis result (examining distributions of action-value coefficients after doubling or quadrupling their angles to examine how relative strengths of different value signals vary across different brain regions; Figure 6 in the revised manuscript). With respect to the possibility of chosen value signals before action selection, we obtained essentially the same conclusions with and without including chosen value in the regression model assessing action-value signals, which is consistent with our previous finding that chosen-value signals are generally weak before action selection. As suggested, we show the results obtained with the model including chosen value (model 3, Equation 5) in the revised manuscript.

[Editors' note: further revisions were suggested prior to acceptance, as described below.]

I have written a summary of our opinion after the discussion directly below here. I have also left the reviews below for emphasis and detail, but please don't feel that you need to address all the points in the reviews. If you can address the central issues in the summary directly below here, we will be happy.This revision has raised some complications in the reviewers' minds that have led to a lot of discussion. In brief, we are not happy with the 2-stage approach that does not lead to p-values for individual neurons that can be trusted.Whilst we agree that this approach goes some way to rebutting the EDL finding, it is a narrow rebuttal which does not provide a good way forward for scientists faced with similar problems in the future. The combination of GLS modelling that does not accurately deal with the autocorrelations, with non-standard application of circular permutations to demonstrate control performance is, in our view, dangerous, and not useful to the community. We are not keen on publishing such an approach in eLife.If the GLS approach does not lead to good corrections for autocorrelations, then we think it incumbent upon you guys to find a non-parametric approach that does.

We fully agree that a better method to evaluate the statistical significance of value-signals at the level of individual neurons was needed in order to firmly establish the extent to which neurons in different anatomical areas encode value signals throughout the brain. Thus, we very much appreciate the reviewer’s effort to provide us with a set of sample codes and illustrate how more appropriate analyses should be carried out. We have followed these suggestions closely in the revised manuscript. Specifically, we evaluated the extent of false positivity for several different methods by using actual behavioral and simulated neural data while maintaining the same distribution of firing rates and AR(1) coefficients as in the actual neural data. In the revised manuscript, we accordingly report the results from the four procedures that yield chance-level rate of false positivity (~5%) for action-value neurons (Figure 2 of the revised manuscript) when tested using the simulated null data. Two of these methods are based on resampling of behavioral data, and include ‘session permutation’ proposed by the previous paper published in *eLife* by Elber-Dorozko and Loewenstein (2018) and the ‘pseudosession’ method proposed in a recent bioRxiv paper (Harris, 2020). The other two are based on Fourier phase randomization of neural data suggested by the reviewer, namely the conventional Fourier phase randomization (FPR) method and a modified version of the amplitude adjusted Fourier transformation method (Theiler 1992 Physica D). Using these 4 new methods, we still found significant action-value and chosen-value signals in multiple areas of the rat brain (Figure 3 of the revised manuscript). Therefore, we believe that our findings robustly demonstrate the presence of action-value signals in the brain.

You can see in the reviews below, that the reviewers also remain concerned about the distinction in the second half of the paper between action values, chosen values, policies etc, which interacts heavily with questions about the linearity of neuronal responses. We remain concerned that there is a danger this will confuse more than clarify the issue for the community. However, we realise that there is a similar section in the EDL paper, and that you guys need to address this. This may be TB's fault (along with the original reviewers) for not flagging this in the original paper. We would appreciate a clear statement in this section in the paper that states that this section is a narrow rebuttal of the EDL paper, and discusses the difficulties in differentiating policy from action values etc. (see reviews below).

We also agree with the reviewer that approaches based on a simple linear regression model cannot adequately be used to classify individual neurons coding different types of value signals, such as action values, policy, and chosen values. Therefore, as suggested by the reviewers and editor, we shortened this section significantly, but kept it as a rebuttal to the previous *eLife* paper published by Elber-Dorozko and Loewenstein. In particular, we added the following text in the revised manuscript: “In reinforcement learning theory, action values are monotonically related to the probability of choosing the corresponding actions, referred to as policy, making it hard to distinguish the neural activity related to either of these quantities. In addition, the activity of individual neurons is likely to encode multiple variables simultaneously (Rigotti et al., 2013). Despite these difficulties, it has been argued that neural signals related to action value might actually represent policy exclusively (Elber-Dorozko and Loewenstein, 2018). To address this issue quantitatively, using the difference in action values (∆Q) and their sum (ΣQ) as proxies for policy and state value, respectively, we tested how signals for action values, policy and state value are related in a population of neurons in different brain structures.”